# Therapist Voices on a Youth Mental Health Pilot: Responsiveness to Diversity and Therapy Modality

**DOI:** 10.3390/ijerph20031834

**Published:** 2023-01-19

**Authors:** Fiona Mathieson, Sue Garrett, Maria Stubbe, Jo Hilder, Rachel Tester, Dasha Fedchuk, Abby Dunlop, Anthony Dowell

**Affiliations:** 1Department of Psychological Medicine, University of Otago, School of Medicine and Health Sciences, Wellington 6242, New Zealand; 2Department of Primary Health Care and General Practice, University of Otago, School of Medicine and Health Sciences, Wellington 6242, New Zealand; 3Department of Psychology, Massey University Wellington, Wellington 6140, New Zealand

**Keywords:** youth mental health, service delivery, diversity

## Abstract

This article explores therapists’ views on a large youth mental health pilot project (for 18–25-year-olds), which included an individual cognitive behavior therapy (CBT)-informed individual therapy component. Therapists’ views on cultural responsiveness, therapy (delivery, modality and duration) and working with LGBTQIA+ youth were explored using two surveys, individual interviews and focus groups at various stages of the life of the pilot. Some therapists saw the CBT approach as imposed on them, preferring familiar therapy modalities. Many therapists were positive toward CBT for its client-centered approach and reported using CBT-informed approaches with many of their clients to good effect. Some therapists felt pressured by their workplace to see clients for fewer sessions than they needed. Therapists wanted to see a more culturally diverse workforce, to increase their cultural competence through training and to have more easily available cultural supervision. There was some acknowledgement of the importance of training therapists to work competently with LGBTQIA+ young people. Involving therapists in co-design of services from the outset will likely benefit future service development.

## 1. Introduction

The mental health of young people is a global public health challenge [1,2] and has been described as ‘mental health’s new frontier’ [3]. Although some action has been taken internationally to promote the implementation of services for young people, mental health needs during this critical period of development are still largely unmet [4]. There is an increasing need for primary mental health services for young people in New Zealand, particularly among marginalized groups, including low income, Māori (indigenous New Zealanders), Pacific Island (Pasifika) and LGBTQIA+ communities [5,6]. Intersectionality, which is the idea that race, class, gender and other individual characteristics intersect and overlap with each other, impacting the lived experience of discrimination [7], likely affects access to mental health services. International research shows the peak period for mental illness onset is adolescence and early adulthood [8] and the mental health needs of young people have increased with the impact of COVID-19 [9]. Through mental health interventions for children and adolescents in the UK children and young people, the Increasing Access to Psychological Therapies (CYP IAPT) program has found promising evidence of benefits from cognitive behavior therapy (CBT)-based therapies, but demand has been so high that there are recommendations that novel, low intensity interventions be developed, along with increased training of therapists [10]. Canadian researchers in the youth mental health field have outlined a set of guiding principles and objectives for initiatives, which include early intervention, vocational outcomes and co-design of services with young people [11].

This article describes therapist experiences of Piki (Piki is a Māori word meaning both to support or aid and to climb or ascend), a youth mental health pilot project in the Wellington region of New Zealand. In particular, this article explores therapists’ views on cultural diversity and responsiveness, LGBTQIA+ responsiveness and perspectives on the ‘fit’ of the CBT-informed therapy approach taken in Piki.

In this paper, we describe three high level themes (with sub-themes) which we chose to focus on from the qualitative data collected from Piki therapists: cultural responsiveness, rainbow (LGBTQIA+), and perspectives on therapy (delivery, modality and duration) for young people’s mental health needs. Other themes, such as therapists’ experience during COVID-19 and their views on telehealth and digital apps are not the focus of the current paper.

## 2. Methods

### 2.1. Piki Background and Main Outcomes

Piki was developed and funded as a service for 18–25-year-olds with mild to moderate mental health needs in the Wellington region of New Zealand (NZ). It was a complex and innovative youth mental health pilot program, with a number of components. Three Primary Health Organizations (PHOs)—organizations responsible for providing management and support functions to a number of primary care practices—two university counselling service partners and a specialist mental health provider worked together to deliver mental health support for young adults. Online wellbeing resources and peer support were also available.

The brief from the funder, the NZ Ministry of Health, was to provide free Cognitive Behavior Therapy (a CBT-based service in line with the successful Increasing Access to Psychological Therapies (IAPT) program in the UK) [12,13]. 

There was ongoing discussion and debate from the outset about therapy content, style and duration for the 18–25-year-old population. The therapy approach evolved from being ‘CBT’ to being ‘CBT-informed’ and included a range of cognitive behavioral therapies: Acceptance and Commitment Therapy (ACT), Focused Acceptance and Commitment Therapy (Fact), Dialectical Behavior Therapy, Behavioral Activation Therapy, Problem-Solving Therapy, Compassion-Focused therapy and Motivational Interviewing, as well as standard CBT. There was pressure from the funder for the pilot to commence as soon as possible, which meant there was a conflation of the design and service rollout phases which limited the ability for meaningful initial co-design with service users and therapists.

Piki was established on a platform of existing services including individual therapy and Puāwaitanga (phone counselling). Additional innovations included intentional peer support (PeerZone), and a digital wellbeing app (Melon). Clients could self-refer through the Piki website as well as being referred by GPs. The intention was that Piki would include a degree of co-design with young people and an important focus was to improve equity of access for traditionally underserved groups including Māori, Pacific, LGBTQIA+, and people experiencing high deprivation.

The individual therapy was provided by therapists based at two universities, three PHOs (including one indigenous health provider) and a youth health service in the Wellington region. Stepped care was provided by clinical psychologists based at a separate organization for clients that was to be referred to when they needed more specialized and/or more intensive therapy. 

Between January 2019 and December 2020, 5479 young people had at least one session of therapy from Piki therapists (excluding Puāwaitanga phone counselling). Services were initially advised that clients could be seen for 12 or more sessions, but in practice most services only offered up to four sessions.

The existing therapist workforce in the partner organizations was supplemented with an additional 20 full-time equivalent staff funded by Piki from the beginning of 2019. The professional and training backgrounds of individual providers varied. They included: counsellors, mental health nurses, social workers, psychologists, occupational therapists, alcohol and drug counsellors and psychotherapists. Just over 20% of therapists did not have existing CBT training, though some received it during the Piki pilot. The four psychologists providing the stepped care were all CBT-trained and acted as supervisors for therapists who completed CBT training during the life of the project. 

The evaluation of Piki (the authors of this paper were on the evaluation team) concluded that overall it was a successful pilot of an innovative integrated mental health service for young people. The peer support service showed great promise (despite being used at a much lower rate than individual therapy—largely because it received a much lower proportion of the available funding and consequently was not widely publicized). The target of approximately 4500 clients per year accessing the Piki service overall was not reached however, nor were targets for delivery to priority groups (Māori, Pacific and high deprivation groups). Although there was a degree of co-design of Piki with young people, with efforts made to seek the input of youth/service users throughout the pilot, in several areas the degree of co-design was limited and/or feedback was not fully utilized. There was limited evidence of support for a dedicated digital app component to the service and it was not widely used by Piki clients. Puāwaitanga phone counselling was only used by 3.5% of clients [14]. 

### 2.2. Therapist Training

A substantial CBT training course was provided to 25 Piki therapists by the University of Otago Wellington. This utilized an existing post-graduate course which was adapted in terms of condensed delivery of workshop content over a shortened (five months), inclusion of increased youth-focused content, a two-hour rainbow training session and a half-day workshop on adapting CBT for Māori. Therapist competence was assessed based on three written assignments comprising a case history, formulation and treatment plan along with videos of sessions with the client who was the subject of the written assignment. Videos were assessed on a structured rating scale, the Cognitive Therapy Scale-Revised (CTS-R) [15] and there was a final oral exam.

In addition, a two-day CBT ‘fidelity training’ course was provided to 11 Piki therapists with pre-existing CBT training, because therapist ‘drift’ away from adherence to specific therapy techniques is a well-known phenomenon [16]. This covered CBT structure, process and models for anxiety and depression and adherence to CBT was assessed (using the CTS-R) using before and after videos of CBT sessions. 

Additional Piki therapist training was offered following a meeting in mid-2019 with therapists regarding their perceived training needs. Training included rainbow (LGBTQIA+) competency training (three sessions in 2019; one session in 2020); Pasifika cultural competency training (one session in 2019 and one in 2020); Māori cultural competency (September 2021) and sexual harm training (November 2020). The therapists also received training in telehealth delivery to prepare them for online therapy delivery during the COVID-19 lockdown. 

### 2.3. Data for This Article

As part of a wider data collection process for the evaluation of the Piki pilot service as a whole, data were gathered from Piki therapists through two different surveys at different time points and through individual interviews and a focus group discussion. All participants provided written consent for their data to be collected and were assured that the reporting of their data would be de-identified. 

### 2.4. Surveys

Survey One was designed by a PHO psychologist and was distributed in mid-2019 to all Piki therapists employed at that time (n = 33). Survey questions aimed to assess therapy experience and practice. Twenty-seven therapists responded (response rate = 82%). See Appendix A for the list of survey questions.

Survey Two was developed by the University of Otago Wellington’s Piki evaluation team and administered in early 2020. The survey aimed to explore the extent of use of standard CBT in therapy, based on dimensions of the CTS-R [13] (Blackburn et al., 2001), as well as questions about role title, hours per week in the Piki role, number of clients being seen, and administrative parts of the job (See Appendix B for the list of survey questions). Due to the emphasis on CBT skill use, the survey was only sent to those who were either: (i) enrolled on the 2019 or 2020 CBT course, or (ii) had completed the Fidelity course, or (iii) were known by the evaluation team to be highly trained and experienced in the use of CBT. The psychologist and evaluation team member who runs the CBT courses sent an email link inviting 41 therapists (who met the above criteria) to complete the survey, with a follow up reminder email a month later. Of the 41 therapists, 37 responded (90% response rate). See Appendix B for the survey questions.

### 2.5. Focus Group/Individual Interviews

In late 2020, all therapists working in the service at the time (n = 20) were invited to participate in a focus group through an email to their team leader (with request to forward on to individuals). No incentive or remuneration was offered. One focus group was subsequently conducted with seven Piki therapists from three of the six service provider organizations. To ensure representation from all organizations, individual interviews were conducted with a further five therapists. The focus group lasted 90 min and interviews ranged from 40 to 60 min. 

There were four question zones with an additional optional zone if time permitted: (1) background to becoming involved in the service; (2) perceptions of the service including the “fit” of the model and training and supervision; (3) relationship between peer support and therapy; (4) perceptions of the benefit of the service to clients; and optional (time permitting) thoughts on the design process of the survey. The data used in this study are part of a larger data set gathered for the evaluation, which explains why the participant identifiers are numbers that go into the 100s.

### 2.6. Data Analysis

Response frequencies were tabulated with number and percentages calculated for all Likert-type survey items, using the number of participants who answered the question as the denominator.

Survey free text data were analyzed using template analysis [17]. Responses to each question in the surveys were initially coded separately (by SG). There was a lot of overlap between codes across the questions, and so second stage coding involved creating themes across questions. Final coding involved forming a combined dataset of themes. Quotes are presented with selected demographic information for each respondent (unique ID, gender, ethnicity, role, and service type).

All interviews and the focus group were audio recorded and transcribed by an independent transcriber. These transcripts were imported into an NVivo 11 file and coded by one researcher (JH) using a content framework as well as inductive thematic codes that were developed by the evaluation team for the purposes of the wider evaluation. The coding relating to the themes selected for the focus of this article was reviewed by a second researcher (FM) to refine and reduce the categories into provisional themes.

These themes were then integrated with those from the survey responses and reviewed and discussed by all members of the research team until consensus on final overarching themes was reached.

## 3. Results

In terms of demographic characteristics, the therapists surveyed and interviewed came from a range of professional backgrounds in the mental health field. They were predominantly female and of European ethnicity, with a wide range of ages (Table 1). The lack of cultural diversity in survey and interview respondents is consistent with the data on all Piki therapists: between November–December 2020, 30/39 identified as female and 9/39 identified as male. The number of Māori in the Piki therapist workforce remained low (5 out of 39 therapists in early 2021), whereas 28/39 identified as European, 3/39 as Pacific and 3/39 as other or unknown ethnicities. The four stepped care psychologists (two male and two female) were all from European/NZE backgrounds [13]. Data as to the gender identity and sexual orientation of the therapists were not collected, which may have proved useful for understanding therapist views but was not within the original brief of the project.

### 3.1. Demographic Characteristics of Respondents

Each theme in the sections below is contextualized at the start with relevant findings from the main Piki Evaluation report. NZE in the identifiers means New Zealand European. For focus group quotes, only the organization of the participant is noted.

### 3.2. Perspectives on Therapy Delivery and Modality

Although there was a strong core of CBT-informed provision within Piki therapy services, the Piki evaluation found considerable variation between services and therapists in the extent to which CBT-informed therapies were delivered. Although Piki was intended as a service for mild to moderate severity, in practice the acuity was often moderate to severe, with most therapists surveyed in January 2020 (88%) reporting seeing at least some clients who were experiencing more than mild/moderate distress. Survey data from therapy clients suggested much of the therapy went well and was appreciated.

Therapists expressed strong views on therapy modality and duration. This section first describes what therapy modalities therapists reported using, then goes on to describe their views on therapy modalities for working with young people.

Survey Two found that the majority (67%) of Piki therapists reported using specific CBT strategies (i.e., setting agendas, using behavioral experiments and homework-setting) with most or all clients. Some (22%) reported using specific CBT strategies with some or about half of their clients. A few reported not using CBT at all (11%). 

There was some therapist concern that the therapy modality and associated CBT training was imposed by the government funder (NZ Ministry of Health), which (not unreasonably) specified CBT as the therapeutic modality, based on the success of the IAPT program in the UK: 

*I think the development of it was definitely driven by the people who were providing the money and not so much by the people who were going to be delivering the therapy and the first thing that comes to mind is the requirement to have CBT as CBT training,…. I understand that they want something that works but making sure that it’s more clinician driven than money driven*.(SPO2_focus group—University provider)

Some therapists expressed preference for using existing therapy modalities, with one asserting that therapists do their best work with modalities they are familiar with:

*…allow people to maybe specify their preferred mode of therapy rather than saying you’ve got to use CBT*.(SPO2_focus group—Stepped Care Provider)

*…therapist connection with client is the best predictor about positive client outcome and so in that case we’re better off as clinicians to use the frameworks that we’re most comfortable with cos we’ll get the best results within that framework, as long as we keep an eye on making a meaningful connection with that client. And so in that respect it might be that CBT only is kind of limiting some therapists to actually, to not do their best work so to speak*.(SPO2_focus group—Stepped Care Provider)

One therapist also expressed concern about the effect of the short timeframe for setting up the Piki pilot, which may have impacted on treatment model fidelity:

*In my opinion it would be useful to have had more time to set up the pilot with clear guidelines including treatment model fidelity*.(Survey 2–23, female, NZE, counsellor, PHO)

In some services, a mismatch was noted between the style of therapy advocated by Piki (CBT-informed, which usually involves four or more regular sessions) and the number of sessions made available to clients. Several therapists expressed concern that, in their service, therapy delivery (modality and duration) was numbers-driven:

*…there was a lot of emphasis on numbers and throughput rather than adhering to a CBT, ACT, motivational interviewing model*.(Survey 2–23, female, NZE, counsellor, PHO)

*Overall, I would like to [see] changes to Piki that enable us as therapists to support clients in achieving positive and meaningful outcomes, rather than the current Piki model which predominantly emphasizes getting numbers through the door*.(Survey 2–12, male, NZE, psychologist, University)

Therapist survey results indicated that the majority felt they were able to see Piki clients for as many sessions as they needed, although 27% reported this as not being the case. Reasons given by therapists for not being able to provide sufficient sessions included system pressure to finish after a few sessions, and high demand for services and resultant delays. Some services, however, did allow an unlimited numbers of sessions. 

Another perceived issue was the variability among services and lack of clarity about the stepped care model whereby clients can ideally access as many or as few sessions as required, as one therapist commented:

*I guess I’d just like to see it sort of somehow more flexible and less flexible at the same time… I think that the boundaries are still a bit blurry and I think young people do like having kind of clear boundaries as much as therapists do*.(Interview 03, female, NZE, mental health nurse, PHO)

Several therapists expressed a strong desire to see clients in line with the usual CBT approach with a series of regular sessions:

*Hold up the principles of the [CBT] training—e.g., plan treatment, regular sessions and review which often becomes overwhelmed by demand*.(Survey 2–2, female, NZE, psychologist, Stepped Care Provider)

*It needs to be re/structured in a way so that clients are able to be seen weekly/fortnightly (at least initially)*.(Survey 2–15, female, NZE, psychologist, PHO)

Several therapists expressed concern that, in their service, there was a tension between service pressures for a low number of sessions per client and therapists’ ability to deliver CBT-informed therapies to the extent they felt their clients needed: 

*…apparent departure from the original therapy model agreed for Piki (e.g., clients being able to access weekly sessions with therapists that follow a CBT model—in particular that is based on an agreed treatment plan/goals and builds on this each week to increase skills and insight/understanding in a structured way*.(Survey 2–21, female, NZE, psychologist, PHO)

*Encouraged to see clients for 1–4 sessions maximum. Told clients often only need one. Told average from British Pilot [IAPT] was three sessions or so. No taking into account there may have been a lot of drop outs or other reasons*.(Survey 2–23, female, NZE, counsellor, PHO)

*The training that we get doesn’t match the service that we deliver at all and I’m actually not really using much CBT because of that*.(Interview 03-female, NZE, nurse, PHO)

Some therapists reported that their clients got better outcomes through having a series of CBT sessions, with some actively resisting service pressures in order to provide regular sessions of CBT over a period of time in order to meet their clients’ needs:

*Since the launch of Piki, my clients who have achieved the best outcomes are the ones who I have worked with by going against the service model (i.e., booked in weekly for 12–20 sessions), whereas making any progress is difficult for clients who can only be seen once a month*.(Survey 2–12, male, European, psychologist, University)

*I see clients weekly or fortnightly, but this is only because I’ve blocked my diary off meaning no one else can book appointments in for me—this is not the norm. I also re-book clients myself directly at the end of each session, but I am aware I am unusual in this respect (I don’t believe this is done by the rest of the team)*.(Survey 2–15, female, NZE, psychologist, PHO)

*Encouraged to do very brief interventions and as a therapist had to push back for ethical, and good practice standards*.(Survey 2–23, female, NZE, counsellor, PHO)

Therapists reported benefits for clients of providing longer duration CBT (6–12+ sessions). 

*I think when the young people engage for six to twelve sessions, they make quite good gains and that they can, they learn some things about themselves, about how they’re thinking, how their thinking impacts how they feel, how that might influence their decision making, their problem solving and their behavior*.(Interview 108, female, NZE, social worker, PHO)

Therapists also liked that CBT is skill-based, has shared formulations and found the worksheets helpful:

*I found that those worksheets and things from the CBT, that that age group just love them, it’s like give me something to take away, give me something so I can see it and understand it and so I’ve just found them probably some of the most useful tools*.(SP02 Focus group–PHO)

*Yeah, clients have really enjoyed sort of formulating, seeing their formulation of their thoughts and behavior on things like whiteboards and stuff. So I have really liked that component of it all*.(SP02 Focus group—University)

Finally, there was a reminder from a therapist that the therapy relationship is at the heart of good CBT as with other therapy modalities:

*CBT is popularized and incredibly useful however do that without empathy, compassion and client front and center it becomes just empty words, choose the practitioners well this work is not possible or probable for all*.(Survey 2–28, male, Pacific, counsellor, PHO)

In summary, some therapists felt that CBT was imposed on them and preferred to work with more familiar therapy modalities; the majority of therapists used CBT regularly with their clients. Many liked CBT for its client-centered approach and reported using CBT-informed approaches with many of their clients to good effect.

### 3.3. Responding to Diversity

In Survey One, most therapists reported considering the inclusion of cultural practices, individual and spiritual preferences, along with whānau (family) involvement, though to varying degrees. Survey Two found there was some use of Kaupapa Māori therapy models by therapists (Kaupapa Māori actively legitimizes and validates Māori language, Māori knowledge and Māori customs).

#### 3.3.1. Need for Cultural Diversity 

Therapists expressed the need for more culturally diverse therapists to be employed, with particular reference to the need for more Māori therapists to be employed.

*I think we need to employ more Māori and Pasifika people. We will need to also focus on needs of Asians*.(Survey 1, male, NZE, social worker, PHO)

*I would say that our Māori clients definitely gravitate towards Māori counsellors*.(Interview 106—female, NZE, phone counsellor, Phone counselling service)

*I definitely have people see me and kind of go, ‘Do you have any Māori colleagues?*.(Interview 03, female, NZE, mental health nurse, PHO)

#### 3.3.2. Concerns about the Fit of CBT for Diverse Groups

Some therapists noted that the model of ‘talking therapy’ and the CBT model favored Pākehā (NZ Europeans), coming, as it does, from a Western tradition:

*I think it’s great if you’re white Europeans. I don’t think it’s great for Māori and Pacific and other, so much*.(Interview 87, female, NZE, team leader, PHO)

*…a lot of it is to do with like talking therapies but not everyone wants to talk and that is like you see with the Pacific boys they don’t always want to talk*.(Interview 43, female, Pacific, youth health nurse, Pacific Health Provider)

*I felt like the structure just wasn’t flowing as well with the Pasifika and Māori clients… that model just didn’t fit with some of my clients coming through within Piki…I’d pull certain parts of CBT, but I couldn’t follow the structured model of the CBT with some of the clients, it just didn’t match*.(Interview 73, female, NZE, counsellor, PHO)

The CBT model was mentioned by one therapist as perhaps less suitable for those who may have lower levels of education, difficult life situations or simply not fitting for that client:

*[for] my ones that weren’t at University—were working, things like that—I found it was a bit harder for them to mould’… ‘those that were I guess had a lot of high needs in the context of life, ones that in general actually just needed to come in and just go ‘wah’, this is happening and life’s really, really… crap right now and has some valid reasons for it… I had kids like they had partners, relationship break ups, things like that…it was just like actually we need to deal with this crisis right now because this is impacting on everything else…like I’d pull certain parts of CBT but I couldn’t follow the structured model of the CBT with some of the clients*.(Interview 73, female, NZE, counsellor, PHO)

#### 3.3.3. Need for Cultural Training

The importance of cultural competence and training was emphasized by some therapists, with a suggestion that this should be compulsory:

*Training in Pacific peoples’ ways of thinking and approaching therapy, Māori tikanga [customs and values] and protocols incorporated into service and specific models of therapy taught and practised with supervision*.(Survey 1, female, NZE, counsellor, University)

*The cultural trainings on offer are fantastic, however they should be made compulsory to ensure all staff are familiar with multicultural services*.(Survey 1, male, European, Psychologist, PHO)

One therapist (in her late 50s) noted that their undergraduate training did not generally prepare them well for working with Māori, although another (early 30s) reported the opposite (which could reflect improvements to cultural content of university courses over time):

*In our Universities through lots of our training, there isn’t too many Māori or Pasifika models taught as therapeutic models and we, certainly we incorporate elements of language and other cultural ideas from other cultures but I’m not sure that they’re as inclusive as they could be*.(Interview 108, female, NZE, social worker, PHO)

*With my undergrad being social work, Te Whare Tapa Whā [Māori health model] and Fonofale [Pacific health model] is kind of embedded within all of our practice, within all of our, everything we do is every year, every paper you’ve got to incorporate what you study, within that*.(Interview 73, female, NZE, social worker, PHO)

#### 3.3.4. Need for Cultural Supervision

Several therapists emphasized the need for more easily accessible cultural supervision:

*More established pathways/relationships to seek cultural supervision and input*.(Survey 1, female, NZE, psychologist, PHO)

*Access to cultural advice/supervisory services for Piki clinicians on a regular/as needed basis*.(Survey 1, female, NZE, psychologist, Stepped Care Provider)

*[We need] cultural advisors that spend time across the teams, so they are visible, accessible, and modelling behavior’*.(Survey 1, female, European, social worker, PHO)

#### 3.3.5. Inclusion of Māori Models

Some therapists reflected on the need for inclusion of Māori models in their practice alongside the CBT model, with Māori clients:

*…like [for] a lot of young people, I often explain the five part model [CBT model] to them on the whiteboard and they often, they go, “Oh right, that’s what I do,” and the lights sort of go on and I think there’s definitely some of that’s very useful but I also think that other models, whether it’s like a Māori model like Te Whare Tapa Whā or maybe, or Te Wheke… some other models could be equally useful*.(Interview female, 108, NZE, social worker, PHO)

#### 3.3.6. Responsiveness to Diversity in Service Delivery

Some therapists were keenly aware of the need to improve the way they work with Māori but felt constrained by existing systems:

*I own that that I probably need to learn more… we’re not really very well set up for, even physically set up, I mean it’s a tiny cupboard of a room I’m in, set up for whānau meetings or… going out into the community to meet with people which might be more appropriate—so it does feel quite constrained… We might have to be a bit more flexible but just the mechanics behind… anything that I might want to do to mix up how I work with Māori clients—I have to work that out*.(Interview 109—female, NZE, Psychologist, Stepped Care Provider)

The fit of the service for the higher needs of some Māori was also questioned by one therapist who worked for the service that more complex clients could be ‘stepped up’ to:

*I think some of the challenges I can see… we come up with this brevity, you’re looking at intergenerational trauma and really, really complex family histories and in the back of my mind I’m thinking—I’ve got twelve sessions and that even opening that box may not be the most appropriate thing*.(Interview 109—female, NZE, Psychologist, Stepped Care Provider)

Several suggestions were made by therapists about ways to improve or change service delivery to meet the needs of diverse groups including Māori and rainbow communities, such as inclusion of family in therapy where possible, creating new ways of delivering services such as home visits, groups and peer support. There was particular emphasis on increasing outreach to populations who have difficulty accessing mental health support.

*Currently, I believe Piki mainly increases access to counselling for those who already had little difficulty accessing counselling (e.g., white cisgender straight Pākehā [Europeans] who are predominantly middle class)*.(Survey 2–12, male, European, psychologist, University)

*Services are kind of still siloed with a kind of white mainstream culture, like our service was doing some outreach at a couple of maraes [Māori meeting houses] based in the region. …but I don’t think it’s really targeted Māori and Pasifika like maybe it was hoping to*.(Interview 108, female, NZE, social worker, PHO)

It was also noted that encouraging Māori to access mental health programs is a challenge that even Māori organizations struggle with:

*It’s a bit of a mission… the Māori organizations supporting whānau—they’re still struggling to get clients to come in to see me under the Piki program*.(Interview 89–male, Other ethnicity, Occupational therapist, PHO)

In sum, the main cultural themes expressed by therapists were that they wanted to see a more diverse workforce to meet client needs, particularly Māori, they wanted to increase their cultural competence through training, and they wanted more easily available cultural supervision. Some had concerns about the origins of CBT in Western traditions and thought that it may be less suitable for people from other cultures. 

#### 3.3.7. Rainbow Responsiveness

Apart from some imagery on the Piki website (www.piki.org.nz, accessed on 28 August 2022), there was no specific marketing targeted towards young rainbow (LGBTQIA+) people accessing Piki, despite this being one of the target groups. Rainbow responsiveness was not a topic that was explored as a particular focus in the interviews, but there were nonetheless some comments in this area. Some therapists who were interviewed identified that they had ‘a few’ rainbow young people as clients. Interviewees from university-based health providers noted that they provided more readily accessible mental health support that was well-utilized by rainbow students, compared to interviewees from community services who reported only seeing a few rainbow clients.

*I saw a large number of that sort of population within a University environment and I think Universities are probably quite good at including that population, in a community setting not so much*.(Interview 108, female, NZE, social worker, PHO)

*I think working within the University environment, that access is already pretty well served I think compared to the general population but I think Piki’s another avenue I think particularly for the Rainbow community*.(Interview 02, female, NZE, AOD counsellor, PHO)

One therapist commented on the importance of training in working with rainbow young people.

*I have a client at the moment who’s transgender, I’ve had another young person who was wanting to transition. I think there is definitely being welcoming and accepting… I almost wonder if we need specific, different training*.(Interview 108, female, NZE, social worker, PHO)

Another therapist noted the high quality of the in-service training received.

*There’ve been other trainings around working with Rainbow through [organization], they’ve all been really high quality, so I think the training’s been good’*.(Interview 109, female, NZE, psychologist, Stepped Care Provider)

In sum, rainbow responsiveness was mentioned, but not strongly emphasized by therapists. There was some mention made of the rainbow community in the youth population and the importance of training therapists to work competently with rainbow young people. 

## 4. Discussion

This study, which included survey, interview and focus group data, explored concerns and potential improvements from a range of therapy providers employed as part of the Piki pilot. Suggested improvements included increased cultural diversity in the therapists employed, more cultural training (and supervision) and listed a number of ways in which more culturally appropriate services could be provided. Some expressed doubts about CBT’s suitability for non-European cultures such as Māori. Although many therapists were happy with the number of sessions they could provide, it is concerning that close to a third of therapists felt constrained by the model being employed by their organization and reported being unable to see clients for as many sessions as needed.

Some negative views as to the cultural appropriateness of CBT may reflect misconceptions about CBT, along with a lack of willingness to adapt it to meet clients’ needs. Although indisputably rooted in Western traditions, CBT has been adapted for other cultural groups with good effect [18], including Māori [19], and CBT training emphasizes the need for cultural sensitivity [20,21] and the incorporation of culturally appropriate models in formulation, such as the Te Whare Tapa Whā model with Māori [22]. CBT has also been adapted for working with rainbow communities in an authentic way, given that it stems from patriarchal, heteronormative assumptions [23], and the rainbow training and CBT training provided to therapists took this into account.

The IAPT approach taken in the UK has its critics: the tendering process and targets for numbers of clients seen can lead to managers limiting the number of sessions, thus risking a reduction in quality of clinical care; therapists may at times apply CBT instead of referring to other services (such as providing CBT for anxiety about debt, instead of referring the client to a debt advisor) and excluding more recently developed ‘third wave’ CBT-based therapies. Further, IAPT has been using an underlying (problematic) paradigm of traditional, medical/ technical models of therapy that assume mental health problems arise from faulty or abnormal mechanisms or processes within the individual and are not context-dependent [24,25]. Although the Piki approach was CBT-informed (and is inclusive of ‘third wave’ therapies) unlike IAPT, some similar concerns were echoed by Piki therapists, particularly with regard to systemic impacts on the number of sessions. 

The compressed timeframes limited providers’ ability to engage with Māori and Pacific networks in co-design, limited the ability of services to recruit from a diverse range of cultures and made extra pre-training of therapists in cultural responsiveness, rainbow responsiveness or CBT-informed therapies impossible. Some therapist comments suggest that Piki service delivery may not be entirely fit-for-purpose for non-European cultural groups and the rainbow community.

A combined service user and therapist focus group for the purpose of co-producing and brainstorming what might work best for young people could have been beneficial in the early stages of the project. Earlier engagement with therapists around what therapeutic approach would likely be the best ‘fit’ for young people and within existing services could have assisted buy-in to whatever therapy approach is used. If CBT is the chosen therapy modality for similar youth mental health interventions, there needs to be an opportunity to clarify misconceptions about it. 

Although it is to be expected that some training needs will emerge during the course of a large pilot such as this, it would have been preferable for the majority of training needs to be identified and addressed before the project commenced, rather than ‘building the plane while flying it’. Nevertheless, the CBT training was well-received and optional extra training was available to therapists that was relevant to their needs, especially in the areas of cultural training and working with rainbow young people. Although the cultural training was well-received, there was an evident need for more cultural supervision, with therapists being uncertain on how to tailor CBT to these populations. Future services would benefit from ensuring that rainbow training, cultural training and resources for how to adapt CBT for other populations (such as those with lower education) is not optional and is conducted prior to the commencement of the service.

The importance of the rainbow training aligns with comments made in interviews with service users who noted that in order for rainbow people to feel welcome, there needs to be specific indications that a provider is supportive and inclusive such as use of appropriately gendered language [14]. Rainbow communities experience high rates of stress, distress and suicide risk [26,27,28] and tend to be under-served by mental health services. One study found that ¼ staff do not feel confident in responding to the needs of rainbow clients and 72% had not received rainbow training [29]. Rainbow clients find disclosing their identity in mental health settings anxiety provoking, fearing that their therapist will lack understanding or respond negatively [30,31,32]. There may therefore be a gap between the perceived responsiveness noted by therapists in this study and the experience of rainbow service users. Therapists who were less competent in this area may not have even been aware that this was an issue. Some therapists used distancing language (such as ‘that sort of population’) when talking about rainbow clients, which suggests some therapists may not have been as responsive as they thought. 

Sessions in Māori and Pacific cultural and rainbow competency training provided opportunities for Piki practitioners to increase their skills and confidence in working with these groups. The therapists were predominantly female and European, and it is important to keep building an appropriately diverse workforce in terms of both cultural and gender diversity, with well-funded training courses and ensuring sufficient run-in time. Increasing cultural diversity of therapists is a work-in-progress and is likely to be impacted by multiple factors including the effects of privilege on the likelihood of completing tertiary education and cultural conceptualizations of mental health, which could influence whether people move into working in mental health settings. Recent advances in research into adapting interventions such as CBT for a range of cultures [18], along with novel therapeutic approaches currently being developed, and development with and by indigenous groups [33,34,35,36,37], may draw more culturally diverse practitioners to the field.

### Strengths and Limitations

This study used surveys and interviews to obtain feedback from therapy providers about a range of issues related to this program. The two surveys provided an opportunity for respondents to provide anonymous feedback about a number of different topics. Survey Two and the interview/focus groups were conducted by the evaluation team who were independent to the therapists’ employers, therefore allowing for more free and frank discussion. Data collection also covered three different time points during the program and so allowed for feedback at different stages in the development of the project. Piki did evolve throughout the life of the project, so comments made earlier in the data collection period may have had less relevance by the end of the project. The results of this study are based on a limited number of survey or interview comments and do not necessarily reflect the views of all therapists. In hindsight, it would have been useful to collect data on the gender identity and sexual orientation of the therapists, as the absence of this information makes it difficult to interpret the therapists’ comments in this area. 

## 5. Conclusions

Therapists have a unique perspective in that they are ‘on the ground’ interacting with young people. Along with the young people using the services, therapists are important voices in the co-design process of a mental health pilot of this nature. Future services would likely benefit from allowing sufficient time before the project commences for such co-design to meaningfully occur. Intersectionality needs to be considered in terms of therapy modality, therapist training and equity of service provision. There should be national and local discussion about the overall aim of therapy for this age group. Although ‘mental health first aid’ may be suitable for relatively low acuity problems, more intensive therapy is required for more complex or severe problems, where there is the potential to support longer-term behavior change, or to intervene early to prevent negative trajectories. Although Piki has tended to provide shorter-duration therapy, it is important that therapy is evidence-based, tailored to individual and cultural needs, and is not overly driven by workload pressures.

## Figures and Tables

**Table 1 ijerph-20-01834-t001:** Demographic Characteristics of respondents.

	Interviewees/Focus Group Attendees N = 13 November–December 2020	Survey 1—PHO N = 27 Mid 2019	Survey 2—Evaluation Team (ET) N = 37 January 2020
Gender			
Female	10	19	10
Male	3	8	27
Profession			
AOD Counsellor	0	2	0
Counsellor	1	6	9
Nurse/Mental Health Nurse	4	5	6
Clinical Psychologist	2	7	8
Occupational Therapist	0	1	0
Social Worker	4	6	8
Educational Psychologist	1	0	0
Therapist	1	0	0
Mental Health Practitioner	0	0	6
Ethnicity			
NZ Māori	0	1	3
Not answered	0	1	0
European (including NZE)	10	25	32
Asian	1	0	0
Other	1	0	0
Pacific	1	0	2
Age		Not collected	Not collected
20–29	7		
30–39	4		
40–49	2		
50+	4		
Organization Type			
Primary Health Organization	5	21	28
Māori Health Provider	0	0	2
University	3	4	3
Stepped Care Provider	2	2	3
Phone Counselling Service	1	0	0
Wellbeing App Service	1	0	0
Not Stated	0	0	1
Pacific Health Provider	1	0	0

## Data Availability

The datasets analyzed during the current study are not publicly available due to confidentiality. We do not have permission for data sharing. The Piki Final evaluation report is available on the University of Otago Website (https://ourarchive.otago.ac.nz/handle/10523/12917, accessed 19 January 2023). This study’s design was not pre-registered.

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
