# Peer review of "Therapist Voices on a Youth Mental Health Pilot: Responsiveness to Diversity and Therapy Modality"

_ijerph, 2023, doi:10.3390/ijerph20031834_

Round 1
Reviewer 1 Report
The paper presents the results of the evaluation by therapists of a pilot mental health program.
I think it is a topic of interest that is well suited to the contents of the journal.
One of the strengths of the paper is the triangulation of methodologies. This adds additional value to the research.
Here are some issues that I think authors should reconsider.
1. Structure of the paper.
In the introduction section, the authors describe the entire Piki program.
I think much of the information provided here should be relocated to the method section.
Instead, in the introduction, I think that information should appear regarding other previous studies similar to the one presented in the paper.
Emphasis should also be placed on theoretical arguments that highlight the importance of intervention programs such as the one described in the paper and their special relevance for populations at risk (such as those discussed in the paper).
I think this is a relevant question that implies a substantial change in the structure of the paper.
2. In the "data analysis" section, the procedure for analyzing Likert-type items should be mentioned. At this time, only how the qualitative content is analyzed is explained.
3. Results.
3.1. In the results section, the report by Dowell et al.
I think that in the results section it is better not to include bibliographical references. This can be confusing.
Although it is a reference to a report on the same investigation, I think it is better to avoid it.
3.2. Regarding the writing style of this section, I think there is a little abuse of the use of literal quotations from the answers of the participants.
The text percentage of these verbatim quotes is too high, compared to the text produced by the authors.
I think that the use of both types of text (literal citations and text from the authors explaining these results) should be more balanced.
4. Formal aspects:
4.1. It should be checked that the first time acronyms appear (for example "CBT", "ACT" or "PHO"), their meaning is specified. This must be done in the abstract, in the main text and in the appendixes.
Although this appears at the end of the paper, I think that specifying it the first time it appears would help the reading.
4.2. Table 1. In the column "Interviewees/focus group attendees" appears n= 12. Should it be n= 13?
Author Response
Response to reviewer
We thank the reviewer very much for their thoughtful and helpful comments.
Reviewer 1 |
Response to reviewer |
1.0 Structure of the paper. |
|
1.1 In the introduction section, the authors describe the entire Piki program. I think much of the information provided here should be relocated to the method section.
|
Thank you for this suggestion. We have relocated the bulk of this information to the methods section as suggested (now sits at 2.1 and 2.2)
|
1.2 Instead, in the introduction, I think that information should appear regarding other previous studies similar to the one presented in the paper. |
We have added the following sections regarding relevant studies to the introduction: Mental health interventions with children and adolescents in the UK CYP IAPT) have found promising evidence of benefits from CBT-based therapies, but demand has been so high that novel, low intensity interventions and being developed, along with increased training of therapists (Ludlow, Hurn & Lansdell, 2020).Canadian researchers in the youth mental health field have outlined a set of guiding principles and objectives for initiatives, which include early intervention, vocational outcomes and co-design of services with young people [11].
While some action has been taken internationally to promote the implementation of services for young people, mental health needs during this critical period of development are still largely unmet [4].
We have also added several other references to international literature to the introduction |
1.3 Emphasis should also be placed on theoretical arguments that highlight the importance of intervention programs such as the one described in the paper and their special relevance for populations at risk (such as those discussed in the paper). I think this is a relevant question that implies a substantial change in the structure of the paper. |
We have added reference to intersectionality theory in the introduction and discussion. In the first paragraph of the introduction we added: Intersectionality, which is the idea that race, class, gender and other individual characteristics intersect and overlap with eachother, impacting the lived experience of discrimination (Crenshaw, 2017), likely affects access to mental health services. And in the discussion we added: Intersectionality needs to be considered in terms of therapy modality, therapist training and equity of service provision. |
2.0 Data Analysis |
|
2.1 In the "data analysis" section, the procedure for analyzing Likert-type items should be mentioned. At this time, only how the qualitative content is analyzed is explained. |
We have added the following explanation: Response frequencies were tabulated with number and percentages calculated for all likert type survey items, using the number of participants who answered the question as the denominator).
|
3. Results. |
|
3.1. In the results section, the report by Dowell et al. I think that in the results section it is better not to include bibliographical references. This can be confusing. Although it is a reference to a report on the same investigation, I think it is better to avoid it. |
We have removed the references from this section as requested. |
3.2. Regarding the writing style of this section, I think there is a little abuse of the use of literal quotations from the answers of the participants. The text percentage of these verbatim quotes is too high, compared to the text produced by the authors. I think that the use of both types of text (literal citations and text from the authors explaining these results) should be more balanced. |
Thank you for this suggestion. We agree that we needed to remove some quotes in order to create more balance. We have removed ten quotes (indicated in track changes) accordingly. Three of these (section 3.3.6) were instead listed as a summary as follows: Several suggestions were made by therapists about ways to improve or change service delivery to meet the needs of diverse groups including Māori and Rainbow communities, such as inclusion of family in therapy where possible, creating new ways of delivering services such as home visits, groups and peer support. There was particular emphasis on increasing outreach to populations who have difficulty accessing mental health support. We are happy to remove some more if the reviewer wants us to. |
4. Formal aspects:
|
|
4.1. It should be checked that the first time acronyms appear (for example "CBT", "ACT" or "PHO"), their meaning is specified. This must be done in the abstract, in the main text and in the appendixes. Although this appears at the end of the paper, I think that specifying it the first time it appears would help the reading.
|
We agree this would be clearer. These changes have been made with the exception of spelling out several psychological tests mentioned in the appendix, which are not substantive to the paper. |
4.2. Table 1. In the column "Interviewees/focus group attendees" appears n= 12. Should it be n= 13?
|
Yes it should be 13. Thank you for pointing out this error. This has been corrected. |
Reviewer 2 Report
A good paper and thank you for the opportunity to review it. There are some points I would like to raise. Firstly, in relation to LGBTIQ, the literature shows that self-declaration of therapist's sexual ID improves engagement in psychological therapies. I don't understand why this is left out.
Also, there is no mention of heteronormative expectations of the psychological therapies framework and if this needs removing? LGBTIQ need to have a gaynormative framework to work within.
And finally, the well-established intersectionality theory (Crenshaw 1983) is not mentioned as how the intersects play a part in diverse groups. This should be added to link in with another expert across the pond.
Author Response
Reviewer 2
We thank the reviewer very much for their thoughtful and helpful comments
Reviewer 2 |
Response to reviewer |
Firstly, in relation to LGBTIQ, the literature shows that self-declaration of therapist's sexual ID improves engagement in psychological therapies. I don't understand why this is left out.
|
We are unclear whether this point refers to therapists self-declaring their sexual ID to the researchers or whether it relates to them self-declaring it to the service users. It may have been useful to have gathered this information from therapists but investigating this area was not within the original brief of the project (although perhaps it should have been). It would have needed careful framing as it is personal information. The benefit of self-declaration of therapist sexual identity for engagement was likely to have been emphasised in the external training provided to therapists. We added the following to the section above Table 1: ‘which may have proved useful context for understanding therapist views but was not within the original brief of the project’.
|
Also, there is no mention of heteronormative expectations of the psychological therapies framework and if this needs removing? LGBTIQ need to have a gaynormative framework to work within.
|
Thanks you for this comment. We agree that conventional therapy stems from western/patriarchal/heteronormative/medical model assumptions and that we need to apply a queer theory lens if we are to work with rainbow communities in an authentic way. The emphasis on ensuring there was extra training in this area for Piki clinicians underscores the importance placed on this. We have accordingly added the following sentence to the discussion in order to emphasise this: CBT has also been adapted for working with rainbow communities in an authentic way, given that it stems from patriarchal, heteronormative assumptions (Martell, 2010) and both the rainbow training and CBT training provided to the therapists took this into account. |
And finally, the well-established intersectionality theory (Crenshaw 1983) is not mentioned as how the intersects play a part in diverse groups. This should be added to link in with another expert across
|
We agree that this is an important and relevant theory to reference given our focus on at risk populations and we know that some intersectionality means mental health services are less accessible. It is also a huge topic that could be a paper in itself and as noted the comments on working with rainbow service users were not a particular focus of the data collected from therapist, but we thought the passing comments made were important to mention – we don’t know what actually happened in sessions with rainbow service users. We have acknowledged this important theory in the first paragraph of our introduction by adding: Intersectionality, which is the idea that race, class, gender and other individual characteristics intersect and overlap with eachother, impacting the lived experience of discrimination (Crenshaw, 2017), likely affects access to mental health services.
And in the discussion we have added: Intersectionality needs to be considered in terms of therapy modality, therapist training and equity of service provision. |
Round 2
Reviewer 1 Report
All the issues that I indicated in my first report have been successfully resolved.
From my point of view, the paper has improved significantly.